# Association between Gastric Cancer and 12 Autoimmune Diseases: A Mendelian Randomization Study

**DOI:** 10.3390/genes14101844

**Published:** 2023-09-23

**Authors:** Qi Wei, Ziyu Wang, Xuanyu Liu, Haibin Liang, Lei Chen

**Affiliations:** 1Department of General Surgery, Xinhua Hospital, Shanghai Jiao Tong University School of Medicine, Shanghai 200092, China; 18895363028@163.com (Q.W.);; 2Department of General Surgery, Jing’an Branch, Huashan Hospital, Fudan University, Shanghai 200060, China; 3Department of Clinical Laboratory, Shanghai Tong Ren Hospital, Shanghai Jiao Tong University School of Medicine, Shanghai 200050, China

**Keywords:** Mendelian randomization, gastric cancer, autoimmune disease, inflammatory bowel disease, causal relationship

## Abstract

Background: Whether the positive associations of gastric cancer (GC) with autoimmune diseases are causal has always been controversial. This study aims to estimate the causal relationship between GC and 12 autoimmune diseases by means of Mendelian randomization (MR) analysis. Methods: After rigorous evaluation, potential candidate single nucleotide polymorphisms (SNPs) for GC and 12 autoimmune diseases were extracted from genome-wide association study (GWAS) datasets. We performed the MR analyses using the inverse variance weighted (IVW) method as the primary approach to the analysis. Three sensitivity analysis methods were added to assess the robustness of the results. In addition, heterogeneity was measured using Cochran’s Q-value, and horizontal pleiotropy was assessed using MR-Egger regression and leave-one-out analysis. Results: The IVW result, which is the main method of analysis, shows no evidence of a causal association between GC and any autoimmune disease. The results of IVW analysis show the relationship between rheumatoid arthritis (*p* = 0.1389), systemic lupus erythematosus (*p* = 0.1122), Crohn‘s disease (*p* = 0.1509), multiple sclerosis (*p* = 0.3944), primary sclerosing cholangitis (*p* = 0.9022), primary biliary cirrhosis (*p* = 0.7776), type 1 diabetes (*p* = 0.9595), ulcerative colitis (*p* = 0.5470), eczema (*p* = 0.3378), asthma (*p* = 0.7436), celiac disease (*p* = 0.4032), and psoriasis (*p* = 0.7622) and GC susceptibility. The same result was obtained with the weighted median and the MR-egger (*p* > 0.05). Conclusion: Our study did not find a genetic causal relationship between susceptibility to these autoimmune diseases and GC, which suggests that unmeasured confounders (e.g., inflammatory processes) or shared genetic architecture may be responsible for the reported epidemiologic associations. Further studies of ancestral diversity are warranted to validate such causal associations.

## 1. Introduction

Gastric cancer (GC) is the fifth most common type of cancer and the third leading cause of cancer death in the world [1]. Although the incidence of GC has declined in the United States and Western Europe in recent decades, it remains a major health problem that cannot be ignored with the significant mortality rate among malignant tumors [2,3,4]. Known risk factors for the condition include infection with *Helicobacter pylori*, a high intake of salt, and a diet low in fruits and vegetables [5,6]. The 5-year survival rate of patients with GC is 95–99% if it is diagnosed at an early, resectable stage and surgery is undergone in a timely manner, while the 5-year survival rate is less than 30% if they are diagnosed at an advanced stage [7,8]. Due to the lack of specific clinical symptoms in the early stage, GC is often diagnosed at an advanced stage, which is associated with worse overall survival [9]. South Korea and Japan have implemented nationwide gastric cancer screening by endoscopy within their National Cancer Screening Program, which has significantly improved the survival of gastric cancer patients [10]. Increasing attention is being paid to the recognition of high-risk patients for GC surveillance. In addition, some immune-mediated diseases have been reported to be associated with an increased risk of GC in previous epidemiologic studies [11].

Autoimmune diseases, which occur in multiple organs or systems throughout the body, are characterized by the loss of self-tolerance, which results in the immune-mediated destruction of the body’s own tissues [12]. Immune dysregulation is thought to play a causative role in the pathogenesis of autoimmune diseases and tumors, and accumulating evidence suggests that autoimmune disorders might be related to the malignancy [13]. An increased incidence of multiple tumors has been found in systemic sclerosis patients with anti-RNAP antibodies, especially in breast and gastrointestinal tumors [14]. Chronic inflammation in rheumatic diseases increases the likelihood of solid malignancies and hematological malignancies. Prolonged inflammation caused by rheumatic diseases is responsible for the transition to secondary malignancies [15]. It has been suggested that gastric neoplasia occurs more in patients with autoimmune gastritis than in the general population [16]. There are also meta-analyses that show a consistent inverse association between pancreatic ductal adenocarcinoma and allergic disease, suggesting that the body’s atopic disease reduces the risk of malignancy [17]. Yet, because of the inherent limitations of observational research, including reporting biases, confounders, and inverse causation, it is not possible for a single study’s results to be used as the sole basis for a clinical trial. Previous studies have failed to clarify the relationship between autoimmune diseases and malignant tumors. The observed association between autoimmunity and gastric cancer remains incidental and requires further validation.

Mendelian randomization (MR) analysis, an emerging statistical method, estimates the causal relationship between exposure and outcome by using genetic variants such as single nucleotide polymorphisms (SNPs) as instrumental variables (IVs) [18]. To reliably predict exposure, MR identifies potential genetic variants as IVs based on the random assortment of genetic variants during meiosis, which is based on the econometric theory for IV analysis. As a result of these characteristics, the effects of confounding and reverse causation can be minimized and the biases typically found in observational studies can be controlled for with MR analysis [19,20]; it has been widely used to investigate associations between exposures and outcomes with excellent precision. It is unclear whether autoimmune diseases have independent effects on GC. In this study, MR analysis was used to investigate genetic causal effects between gastric cancer and 12 autoimmune diseases.

## 2. Methods

### 2.1. Study Design

Figure 1 depicts the flow diagram of the MR study between GC and 12 autoimmune diseases. Three criteria must be met in order to accurately infer the potential causal association between GC and autoimmune disease through the MR approach [21,22]: (1) genetic variants should be significantly related to the exposure; (2) when extracted as instrumental variables for exposure, genetic variants are not related to other confounders; (3) genetic variants have an effect on the outcome of a disease solely through their effect on the exposure to it. There was no need for additional ethical approval because we were using publicly available summary statistics from genome-wide association studies (GWAS).

### 2.2. Study Cohorts and GWAS

We performed a systematic analysis with GWAS summary-level data obtained from different large-scale cohorts to infer causal relationships between GC and autoimmune diseases [23]. GWAS summary statistics on GC (GWAS ID: finn-b-C3_STOMACH_EXALLC), including 633 GC cases and 174,006 controls of European ancestry, were downloaded at https://www.finngen.fi/en/access_results (accessed on 10 June 2023) [24]. The summary statistics of the GWASs for 12 autoimmune diseases (Table 1) were downloaded from the GWAS catalog website (https://gwas.mrcieu.ac.uk/ (accessed on 10 June 2023)). Autoimmune disorders in this study include rheumatoid arthritis (RA), systemic lupus erythematosus (SLE), multiple sclerosis (MS), ulcerative colitis (UC), Crohn‘s disease (CD), celiac disease (CeD), asthma, eczema, type 1 diabetes (T1D), primary sclerosing cholangitis (PSC), and psoriasis (PsO). In these studies, all of the cohorts of cases and controls were of European ancestry, and there is also no significant overlap between the populations of the GWAS. The initial studies provide detailed information on enrollment procedures and diagnostic criteria. More detailed information for the cohorts of cases and controls can be found in Table 1.

### 2.3. IV Selection

We used the R package TwoSampleMR to select genetic instruments from each of the 12 autoimmune disease GWASs [25]. A rigorous quality control process was used to select eligible instrumental variables for each autoimmune disease. We used independent genetic variants that were significantly associated with each exposure (*p* < 5 × 10^−8^) for each instrument and used a clustering procedure with R^2^ < 0.001 and cluster distance = 10,000 kb to avoid linkage disequilibrium (LD) [26]. Then, we excluded all ambiguous or palindromic SNPs that had a nonconcordant allele (for example, *A/G* versus *A/C*) or ambiguous strand (for example, *A/T* or *G/C*). To detect the underlying weak instrumental variable bias, we also calculated the proportion of variance explained (R2) and the F statistic for all SNPs. A mean F-statistic  >  10 indicates suitable instrumental variables [27,28].

### 2.4. Statistical Analysis

In this study, we estimated the causal associations between 12 autoimmune diseases and GC separately using different MR methods by combining summary statistics (β coefficients and standard errors) [18]. We applied three MR analysis methods based on different assumptions, including inverse-variance-weighting (IVW), MR-Egger regression, and weighted mean (WM) [29,30]. If the horizontal pleiotropy is balanced, then the IVW method will provide an unbiased estimate [25]. The multiplicative random effects IVW method, which assumes that all SNPs are valid instruments but provides the most accurate estimates, was used for the primary MR analyses [31]. In addition, an online web tool (https://sb452.shinyapps.io/power/ (accessed on 10 June 2023)) was applied to calculate the statistical power of each IV. The odd ratios (ORs) are described as an increase in the level of the risk factor per standard deviation (SD) [32]. 

### 2.5. Robust Analysis

To detect potential horizontal pleiotropic effects and to examine the consistency of the associations, several sensitivity analysis methods were performed, including weighted median, MR-Egger, and MR pleiotropy residual sum and outlier (MR-PRESSO) analyses [18]. It is well-known that selecting IV SNPs from the exposure GWAS results in the winner’s curse [33]. To reduce the bias of Mendelian randomization analyses when performed, we used sensitivity and multiplicity analyses. The winner’s curse is a relevant consideration in the selection of IVs from the exposure GWAS, but it should not materially bias the estimate or the overall conclusion [34]. Directional horizontal pleiotropy in the causal estimates may be indicated by the intercept term in the MR-Egger regression. Cochran’s Q test was used to test for heterogeneity in the causal estimates between exposure and outcome, and we applied both the estimates for causality of the IVW method with fixed effects and the MR-Egger regression to recognize heterogeneity. Heterogeneity was quantified using Cochran’s Q statistic and a *p* value < 0.05 was considered significant heterogeneity [35,36]. Multipotency was further analyzed using the R package MR-PRESSO to reduce bias (based on the IVW results) and to remove any outliers. For the tool to perform the MR-PRESSO outlier test, at least 50% of the genetic variance must be valid according to the InSIDE hypothesis. [36]. In addition, a leave-one-out analysis was performed to determine the stability of the results to identify potentially heterogeneous SNPs [37].

R tool (version 4.3.0) was used for all statistical analyses in this study. The MR analyses were carried out using the R packages TwosampleMR (version 0.5.6), MR-PRESSO (version 1.0), and qvalue.

## 3. Results

The causal relationship from MR analysis between GC and autoimmune diseases is graphically summarized in Table 2. In the primary IVW MR analysis, we did not find a significant causal association between 12 autoimmune disorders and the risk of GC, nor did the other methods, including MR-Egger regression and penalized weighted median (all *p* > 0.05). The results of IVW analysis show the relationship between RA (*p* = 0.1389, 95% CI = 0.9781–1.1719, OR = 1.0706), SLE (*p* = 0.1122, 95% CI = 0.9875–1.1273, OR = 1.0551), CD (*p* = 0.1509, 95% CI = 0.9761–1.1695, OR = 1.0685), MS (*p* = 0.3944, 95% CI = 0.9353–1.1849, OR = 1.0527), PSC (*p* = 0.9022, 95% CI = 0.9043–1.1209, OR = 1.0068), PBC (*p* = 0.7776, 95% CI = 0.9124–1.1304, OR = 1.0156), T1D (*p* = 0.9595, 95% CI = 0.9399–1.0675, OR = 1.0016), UC (*p* = 0.5470, 95% CI = 0.9209–1.1682, OR = 1.0372), eczema (*p* = 0.3378, 95% CI = 0.6227–1.1765, OR = 0.8559), asthma (*p* = 0.7436, 95% CI = 0.8142–1.3337, OR = 1.0420), CeD (*p* = 0.4032, 95% CI = 0.8242–1.0808, OR = 0.9438), and PsO (*p* = 0.7622, 95% CI = 0.9724–1.0207, OR = 0.9963) and GC susceptibility. F-statistics for PBC and PsO could not be calculated due to missing data from some GWASs. In addition, F-statistic > 62 for all of the instruments, which is above the standard cut-off (>10) and indicates that the instrument has sufficient power.

For each IV, a sensitivity analysis is performed to detect the presence of horizontal pleiotropy. IVs for PBC were found to be significantly heterogeneous (IVW: Cochran’s Q = 0.027). SNPs with heterogeneity were found using MR-PRESSO analysis with R. Then, four SNPs (rs7775055, rs79513546, rs8067378, and rs911263) with heterogeneity were extracted using MR-PRESSO analysis, the SNPs were removed for outliers, and the MR analysis was performed again. The MR-Egger regression intercept was insignificant (Figure 2). Sensitivity analyses were consistent with no evidence of bias due to genetic pleiotropy. Visual inspection of funnel plots (Figure 3) and leave-one-out plots (Figure 4) did not reveal any obvious directional pleiotropy. Due to the lack of effective SNPs for GC in reverse MR studies, reverse MR analysis was not performed. SNPs for the 12 autoimmune diseases included in the study can be found in the Appendix A.

## 4. Discussion

To the best of our knowledge, this is the first study that has used MR analysis and large-scale GWAS data sets to demonstrate a causal relationship between GC and autoimmune diseases. Our results suggest that there is no apparent genetic causal relationship between autoimmune diseases and the chance of GC, which means that there is a lack of genetic association with correlating traits. This finding is consistent across multiple GWAS data sources and has been confirmed in sensitivity analyses.

Recent studies have shown that several autoimmune diseases can cause a slight increase in the risk of developing multiple cancers [38,39]. In addition, a negative association between gastric cancer and some autoimmune diseases has also been observed [40]. Autoimmune diseases may directly or indirectly contribute to gastric cancer through several pathways. The role of chronic inflammation or immunosuppressive drugs in the development of gastric cancer should be considered, as our study confirmed that there is no genetic link between autoimmune diseases and gastric cancer. The effective *Helicobacter pylori* eradication strategies have greatly reduced the incidence of gastric cancer compared to the previous period. Perhaps rational immune management will be a new direction for the reduction of the incidence of gastric cancer or the improvement of the survival rate of gastric cancer.

In contrast to our findings, a study that included 30 different immune system diseases [38] found that type 1 diabetes, systemic lupus erythematosus, and primary biliary cirrhosis significantly increased the risk of GC. However, the result is likely to be biased because data on some exposure factors were missing and there was no evidence of trim and fill correction for missing information. In a study of 4.5 million U.S. male veterans [41], a history of autoimmune disease with localized effects on the digestive tract generally increased the risk of cancer in the organs affected by the autoimmune disorders, such as pernicious anemia and stomach cancer, which does not contradict our results. The relationship between other digestive organs and autoimmune diseases with local effects in the digestive tract needs further investigation.

SLE, RA, and MS are characterized by chronic, systemic, and exaggerated immune activation and inflammation affecting nearly every tissue in the body. Chronic inflammation caused by autoimmune diseases has long been implicated in carcinogenesis, but no genetic causality between these disorders and GC was found in this study. In a large retrospective study, Zhou Z et al. found that RA generally increased the incidence of solid tumors, with no significant association with GC. Most of the tumors developed nine years after the diagnosis of RA, suggesting that the potential carcinogenic effect of long-term use of antirheumatic drugs or NSAIDs deserves attention. Patients with SLE and MS also have a significantly increased risk of cancer, especially cancer of the female reproductive organs and solid cancers, with no significant association with GC [42].

Inflammatory bowel disease plays a significant role in the development and progression of gastrointestinal tumors, and the association between the risk of colorectal cancer and inflammatory bowel disease is well established [43]. However, in our study, we have not been able to find a genetic cause-and-effect relationship between inflammatory bowel disease and GC. Wan et al. confirmed that IBD is significantly associated with an elevated risk of digestive tract tumors. After stratification by cancer location, IBD primarily elevated the chance of colorectal cancer, but not stomach cancer. In addition, Crohn’s disease (CD) significantly increases the risk of small bowel and colorectal cancer, whereas ulcerative colitis (UC) only increases the risk of colorectal cancer [44,45]. Interestingly, the *ATG16L1* T300A variant, a major CD susceptibility allele, has been shown to be associated with the development of gastric cancer susceptibility [45], and is also associated with increased overall survival and tumor apoptosis as well as inhibition of *EGFR* and *PPAR* pathways in gastric cancer [46].

Patients with PSC have been reported to have a higher incidence of cancer than the general population, but the reason for the increased cancer risk is unclear and may be related to chronic liver and intestinal inflammation [47]. In a matched cohort study of 1432 patients with confirmed PSC and 14,437 controls with a mean follow-up of 15.9 years, Båve et al. found that patients with PSC had a higher risk of hepatobiliary and pancreatic cancer than the general population, while the risk of gastric cancer was not increased [48]. Celiac disease is defined as an autoimmune disease in which the small intestine is the primary target, and is thought to be one of the causes of several malignancies. Some large cohort studies have reported that CeD does not increase the risk of stomach cancer [49,50].

We found that genetically predicted eczema and asthma were not causally related to GC risk. However, the link between asthma and cancer has been mixed in previous studies. Two independent longitudinal cohort studies involving 480,637 participants suggest that the development of asthma increases the risk of secondary cancer overall, and this association remained after a stratified study based on age and sex status [51]. There is also evidence that asthma may only be a risk factor for lung cancer, but not for other types of invasive cancer [52]. Interesting to note is that an epidemiologic study found an inverse association between GC and allergic disease. El-Zein M et al. found that allergic diseases caused by an overactive immune system may lead to more effective elimination of abnormal cells, reducing the risk of cancer [40]. A cross-sectional study showed that Helicobacter pylori infection was inversely associated with asthma in people aged <40 years (OR = 0.503; 95% CI = 0.280–0.904, *p* = 0.021) [53]. The reason may be that Helicobacter pylori infection prevents allergic asthma by inducing regulatory T cells [54,55]. Allergic diseases are also negatively correlated with other malignancies such as lung cancer and cutaneous cancer [56,57]. In addition, there is also some evidence that allergic conditions may be a risk factor for some types of hematologic malignancies [58]. Further research is needed to determine the exact mechanisms underlying allergic disease in the development of GC.

Our research has several major strengths, the most important of which is the MR design, which allows causal inference free from confounding and reverse causality [18]. Secondly, to ensure the generalizability of causal associations, we included multiple independent large cohorts for MR analyses and subsequent meta-analyses to ensure a sufficient sample size of the outcome. In addition, we used a variety of supplementary analyses, such as sensitivity analyses for heterogeneity, pleiotropy, and funnel plots, to test the robustness of the assumptions for the instrumental variables. Thus, our results should be reliable. 

Nevertheless, we still have a number of limitations, and therefore the results should be treated cautiously. Firstly, the participants in the study were of European descent, so further research is needed to determine whether the findings apply to other populations. Second, the results may be biased because there is an additional complicating factor in using Finnish genetic data for the results: The Finnish population is one of the most diverse from the European population. Furthermore, there are also limitations to using summary GWAS statistics. It is well-known that selecting IV SNPs from the exposure GWAS results in the winner’s curse. Although sensitivity and multiplicity analyses have been performed to assess the validity of Mendelian randomization studies, this does not completely eliminate the bias it may introduce. Finally, although we have taken steps to eliminate the possibility of confounding, we cannot completely rule out the possible influence of horizontal pleiotropy on our results.

## 5. Conclusions

Our MR results suggest that there is no genetic causal relationship between gastric cancer and 12 autoimmune diseases, which means that there is a lack of genetic association with correlating traits. However, this does not exclude the possibility that they are related to unmeasured confounders (e.g., inflammatory processes) other than genetic. To further dissect the complex relationship between GC and autoimmune disease, more population-based and experimental studies are warranted.

## Figures and Tables

**Figure 1 genes-14-01844-f001:**
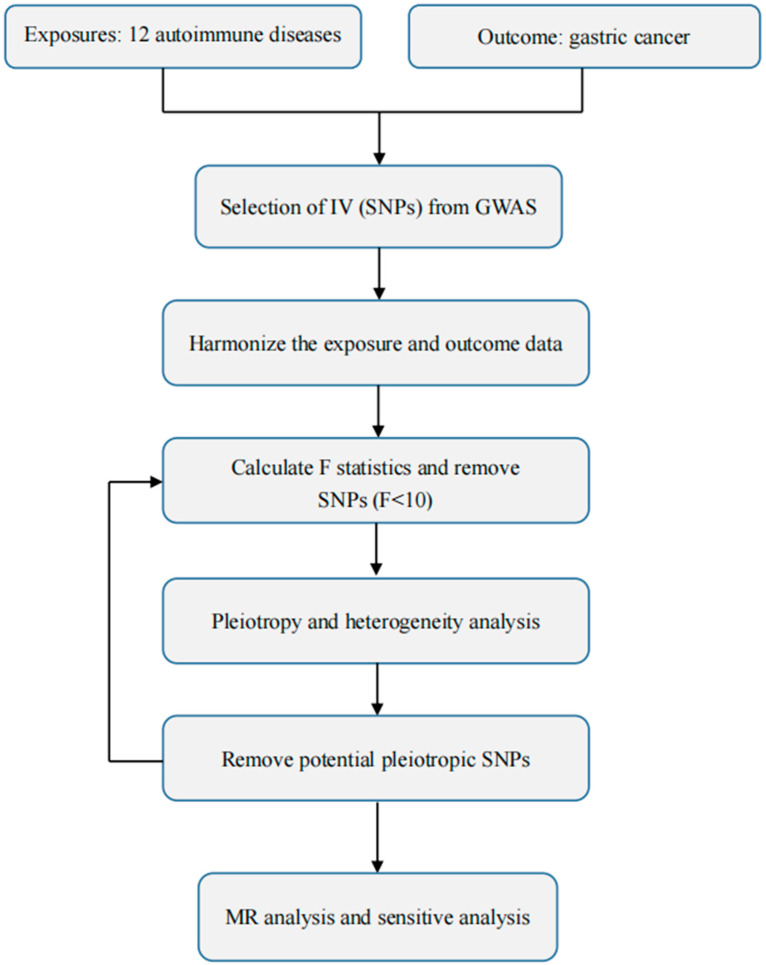
Flowchart of the MR study between GC and 12 autoimmune diseases.

**Figure 2 genes-14-01844-f002:**
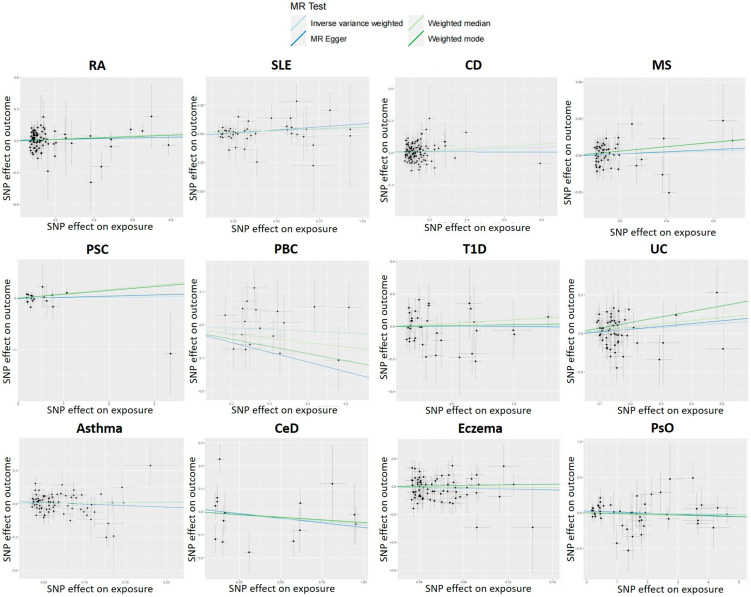
Scatter plots show the MR effect of each exposure on ALS in different MR methods.

**Figure 3 genes-14-01844-f003:**
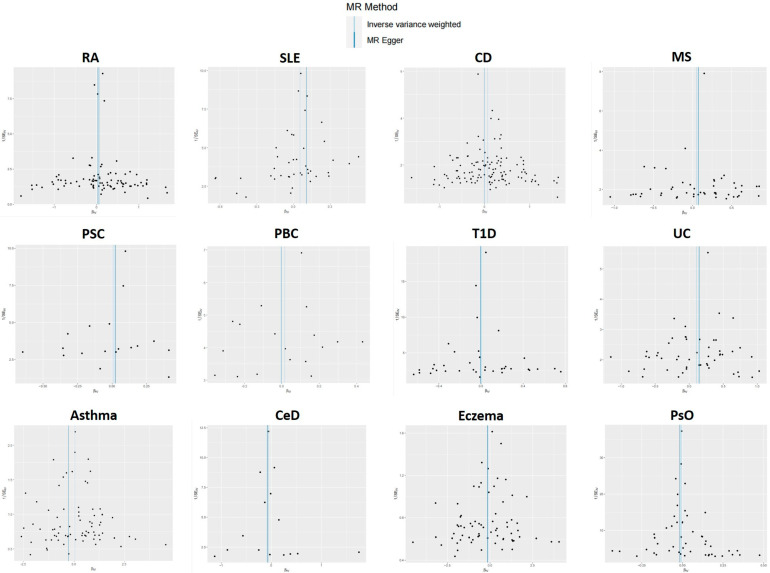
Funnel plots for the causal relationship between autoimmune diseases and GC.

**Figure 4 genes-14-01844-f004:**
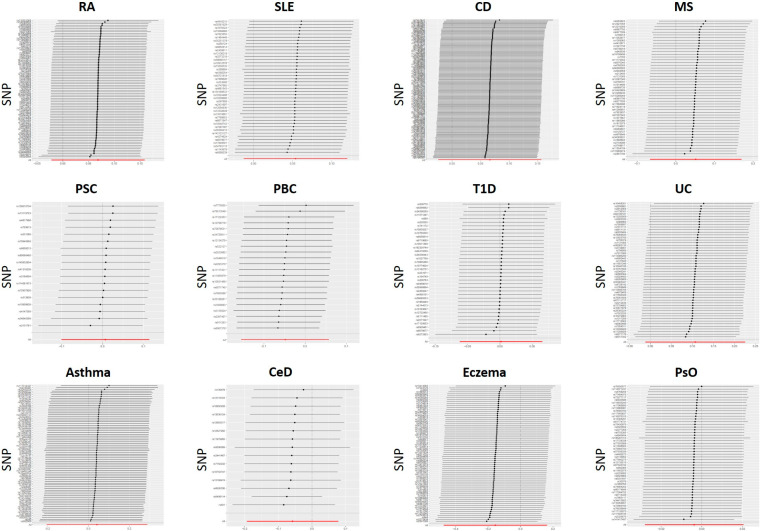
Leave-one-out plots for the causal relationship between autoimmune diseases and GC.

**Table 1 genes-14-01844-t001:** Characteristics of the GC and autoimmune disease GWAS cohorts.

Disease	Study	Journal	Cases	Controls	Sample Size	Datasets in the GWAS
RA	Ha E et al.	*Ann Rheum Dis.*	14,361	43,923	58,284	ebi-a-GCST90013534
SLE	Bentham J et al.	*Nat. Genet.*	5201	9066	14,267	ebi-a-GCST003156
CD	Liu JZ et al.	*Nat. Genet.*	17,897	33,977	51,874	ieu-a-12
MS	Beecham AH et al.	*Nat. Genet.*	14,498	24,091	38,589	ieu-a-1025
PSC	Ji et al.	*Nat. Genet.*	4796	19,955	24,751	ieu-a-1112
PBC	Liu JZ et al.	*Nat. Genet.*	2861	8514	11,375	ebi-a-GCST005581
T1D	Forgetta V et al.	*Diabetes.*	9266	15,574	24,840	ebi-a-GCST010681
UC	Liu JZ et al.	*Nat. Genet.*	13,768	33,977	47,745	ieu-a-970
Eczema	Ferreira MA et al.	*Nat. Genet.*	180,129	180,709	360,838	ebi-a-GCST005038
Asthma	Valette K et al.	*Commun Biol.*	56,167	352,255	408,442	ebi-a-GCST90014325
CeD	Trynka et al.	*Nat. Genet.*	12,041	12,228	24,269	ieu-a-1058
PsO	Tsoi LC et al.	*Nat. Genet.*	10,588	22,806	33,394	ebi-a-GCST005527

RA—rheumatoid arthritis, SLE—systemic lupus erythematosus, CD—Crohn’s disease, MS—multiple sclerosis, PSC—primary sclerosing cholangitis, PBC—primary biliary cirrhosis, T1D—type 1 diabetes, UC—ulcerative colitis, CeD—celiac disease, PsO—psoriasis.

**Table 2 genes-14-01844-t002:** Results of MR analyses between autoimmune disease liability and GC risk.

Exposure	Nsnp	*r* ^2^	*F*	Inverse Variance Weighted			Weighted Median			MR Egger		
				OR	CI	P	OR	CI	P	OR	CI	P
RA	85	0.00187	109.9294	1.0706	0.9781–1.1719	0.1389	1.0819	0.9537–1.2275	0.2211	1.0351	0.9012–1.1891	0.6262
SLE	42	0.00669	76.7697	1.0551	0.9875–1.1273	0.1122	1.0613	0.9672–1.1645	0.2087	1.1121	0.9636–1.2835	0.1540
CD	120	0.00173	90.0585	1.0685	0.9761–1.1695	0.1509	1.1072	0.9611–1.2755	0.1584	0.9974	0.7812–1.2733	0.9831
MS	49	0.0018	69.8125	1.0527	0.9353–1.1849	0.3944	1.1582	0.9547–1.4051	0.1362	1.0764	0.8495–1.3640	0.5450
PSC	18	0.00757	116.2914	1.0068	0.9043–1.1209	0.9022	1.0858	0.9378–1.2572	0.2706	1.0228	0.8451–1.2379	0.8198
PBC	18	NA	NA	1.0156	0.9124–1.1304	0.7776	1.0589	0.9107–1.2311	0.4568	0.9965	0.6733–1.4750	0.9863
T1D	37	0.00456	116.1002	1.0016	0.9399–1.0675	0.9595	1.0421	0.9544–1.1379	0.358	0.9962	0.9047–1.0970	0.9387
UC	85	0.00194	93.2096	1.0372	0.9209–1.1682	0.5470	1.0121	0.8508–1.2040	0.8920	1.1502	0.8587–1.5408	0.3507
Eczema	70	0.000174	62.6772	0.8559	0.6227–1.1765	0.3378	0.9736	0.6202–1.5284	0.9074	0.8697	0.3507–2.1569	0.7641
Asthma	76	0.000168	68.5165	1.0420	0.8142–1.3337	0.7436	1.0629	0.7531–1.5000	0.7286	0.7672	0.4107–1.4339	0.4086
CeD	15	0.0114	285.7935	0.9438	0.8242–1.0808	0.4032	0.9509	0.8416–1.0744	0.419	0.9224	0.7495–1.135	0.4591
PsO	50	NA	NA	0.9963	0.9724–1.0207	0.7622	0.9941	0.9604–1.0291	0.7388	0.9846	0.9554–1.0147	0.3178

ORs express each exposure’s liability impact on GC risk. RA—rheumatoid arthritis, SLE—systemic lupus erythematosus, CD—Crohn‘s disease, MS—multiple sclerosis, PSC—primary sclerosing cholangitis, T1D—type 1 diabetes, UC—ulcerative colitis, CeD—celiac disease, PsO—psoriasis.

## Data Availability

All data involved in this study are available in the public database. For further information, please contact the corresponding authors.

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
