# Peer review of "Association between Gastric Cancer and 12 Autoimmune Diseases: A Mendelian Randomization Study"

_genes, 2023, doi:10.3390/genes14101844_

Round 1

Reviewer 1 Report

Wei et al. have analyzed the genetic causation of 12 autoimmune diseases on gastric cancer through Mendelian Randomization analysis. The work seems sound but I have some concerns that I want to address to the authors. 

Major concerns

1)    One major limitation of the study (slightly mentioned in the discussion) is the selection of the summary statistics. The ones used are the summary statistics ready to use through MRC-IEU and its R package, but more summary statistics from other European populations and East Asian populations can be found at GWAS Catalog, for example. And it is not difficult to process those summary statistics using the R package from MRC-IEU. This is an important point since the authors state that more analyses should be made (lines 265-267) when they can do some additional analyses. You can use different summary statistics of the same autoimmune disease as exposures, and you can use other summary statistics as outcome for gastric cancer (https://www.ebi.ac.uk/gwas/efotraits/MONDO_0001056). In fact, the use of FinnGen data as outcome has an additional complication: Finish population is one of the most divergent from European populations, a critical point that is not discussed. 

2)    In the manuscript it should be clearly stated that there is not genetic causal relationship, since it is what in MR is analyzed. For example, this aspect is mentioned in the conclusion “However, this does not exclude the possibility that they are related at a level other than genetic” (lines 263-264) as it should be in all the manuscript. The causation could exist but not through genetic risk, it could be only a consequence of inflammatory processes. Since the major part of the discussion is based on discussing the evidences of correlation between autoimmune diseases and gastric cancer, it is important to highlight that the genetic link is what is analyzed in this work; and what means that the lack of genetic link in correlated traits.

3)    It is stated that “In these studies, all of the cohorts of cases and controls were of European ancestry, and there is also no significant overlap between the populations of the GWAS” (lines 102-103). In the case of ebi-a-GCST90013534, European and East Asian populations are analyzed. In addition, it should not be any overlap between population, since MR should be done between independent cohorts. Please, check that there is not any overlap or use MR methods that considers the overlap. 

Minor concerns

1)    In the abstract, it is stated that “Our study did not investigate a causal relationship between susceptibility to these autoimmune diseases and GC” (lines 28-29), I guess that it is ““Our study did not find a causal genetic relationship…”

2)    Please, check the names of the genes or Helicobacter pylori to put them in italics.

3)    The ATG16L1 T300A variant is discussed (lines 213-214), that SNP is included as instrument in the analysis of Crohn’s Disease as outcome?

4)    In the instruction of using FinnGen data it is stated that.

When using these results in publications, please remember to: 

Acknowledge the FinnGen study. You can use the following text:

“We want to acknowledge the participants and investigators of the FinnGen study”

Cite our latest publication:

Kurki, M.I., Karjalainen, J., Palta, P. et al. FinnGen provides genetic insights from a well-phenotyped isolated population. Nature 613, 508–518 (2023). https://doi.org/10.1038/s41586-022-05473-8

Furthermore, if possible, include "FinnGen" as a keyword for your publication.

Please, follow those instructions.

Author Response

Dear Reviewers:

Thank you for your comments concerning our manuscript entitled “Association between Gastric Cancer and 12 Autoimmune Dis-eases: A Mendelian Randomization Study” (ID: genes-2575570). These comments are all valuable and very helpful for the revision and improvement of our paper, as well as the important guidance significance for our research. We have carefully reviewed the comments and made corrections that we hope you will agree with. Revised portion are marked in red in the paper. The most important corrections in the paper and the answers to the comments of the reviewer can be found in the appendix.

Reviewer 2 Report

This study concerns the causal effect of 12 autoimmune diseases on gastric cancer. Three popular Mendelian analysis methods are used. The R package twosampleMR is used for all the analyses. That is, the analysis procedure is pretty standard. 

It is well-known that the selecting IV SNPs from the exposure GWAS results in winner's curse. But there is no discussion on this important topic. 

Minor comments:

1. In line 38,  the sentence "Although the incidence of GC has declined in the 38 United States and Western Europe in recent decades, it remains a major health problem 39 that cannot be ignored, particularly in East Asian countries" seems to suggest that the focus of the study is on East Asian population. However, the focus is on population of European ancestry. This sentence can be better organized. 

2. line 83: "genetic variables" should be "genetic variants". 

3. Figure 1: specify which GWAS is used to select the IV SNPs.

4. line 167 and other places: what is "packet analysis" in "MR-PRESSO packet analysis"

The quality of English language is acceptable. 

Author Response

(The authors gave the same response as above.)

Reviewer 3 Report

First of all, I applaud the submission and consideration of this manuscript despite the lack of statistically significant causal associations between the gastric cancer and autoimmune disease markers. Furthermore, the study appears to provide novel information in this area, uses appropriate methods, and is well-described in the manuscript. I have minor suggestions:

“Sensitive analysis” is used in several places where I think you mean “Sensitivity analysis”.

Line 92: “We performed a systematically analysis by GWAS summary…” should be “We performed a systematic analysis with GWAS summary…” (grammar fix). Continuing on line 93, “to draw reliable conclusions about the causal…” should just be “to infer causal…” or something similar (interpretation of whether your conclusions are reliable is up to the reader).

Line 126: please clarify what you mean or give a reference to explain how the IVW method provides the most accurate estimates despite the assumptions.

Line 142: “to ensure accuracy of the results…” should be “to reduce bias…” or something similar (again the judgment call that you have ensured accuracy is up to the reader).

It would be good to show the range of SNP prevalence in the European ancestry population for each autoimmune disease since this might have impacted selection. In general, further discussion of the limitations of using summary GWAS statistics would be warranted even for this audience, besides the European ancestry limitation that you mention. Perhaps a supplemental dataset or table on each study and SNP included would be good to include for those who may want to delve into the nuances further.

Overall nice study.

Very good with a few minor spelling/grammar corrections required.

Author Response

(The authors gave the same response as above.)

Reviewer 4 Report

Thanks for giving me an opportunity to review this paper. In this study, the authors tried to estimate the causal relationship between gastric cancer and 12 autoimmune diseases using Mendelian randomization (MR) analysis, reporting no causal relationship.  This is a good topic but need to improve a lot to make it publishable. Following are my comments-

In abstract:

line 15: "... by means of MR"  MR what?. 

Line 16: "After rigorous evaluation...." What kinds of rigorous evaluation?

Results: all the results show a p-value, is it the most important indicator to judge causal association or any other metric?

Conclusion: "Our study did not investigate causal relationship..." Then what have you investigated? I think it was your study aim.

Introduction 

It is too general. please provide important information related to your study hypothesis.

Methods:

How did you identify autoimmune disease?

robust analysis: The authors did not provide any information about IVW. how they have used this approach.

Discussion:

No clinical implication section.  

Conclusion:

Need to rewrite it again.

Need moderate editing

Author Response

(The authors gave the same response as above.)

Round 2

Reviewer 4 Report

It can be considered for publication.

Minor revision is needed.